# Assessing the Divergent Soil Phosphorus Recovery Strategies in Domesticated and Wild Crops

**DOI:** 10.3390/plants14152296

**Published:** 2025-07-25

**Authors:** Mary M. Dixon, Jorge M. Vivanco

**Affiliations:** Department of Horticulture and Landscape Architecture, Colorado State University, Fort Collins, CO 80523, USA; j.vivanco@colostate.edu

**Keywords:** phosphorus, domestication, rhizosphere, plant-microbe interaction, *pqqC*

## Abstract

Plant-essential phosphorus (P) is a sparingly available mineral in soils. Phosphorus fertilizers—produced by the transformation of insoluble to soluble phosphates—are thus applied to agroecosystems. With advancements in commercial agriculture, crops have been increasingly adapted to grow in fertile environments. Wild crop relatives, however, are adapted to grow in unfertilized soils. In response to these two conditions of P bioavailability (fertilized agroecosystems and unfertilized natural soils), domesticated crops and wild species employ different strategies to grow and develop. It is essential to understand strategies related to P acquisition that may have been lost to domestication, and here we present, for the first time, that across species, modern cultivars engage in physical (i.e., root morphological) mechanisms while their wild relatives promote ecological (i.e., root-microbial) mechanisms. Domesticated crops showcase shallower root system architecture and engage in topsoil foraging to acquire P from the nutrient-stratified environments common to fertilized agroecosystems. Wild species associate with P-cycling bacteria and AM fungi. This divergence in P recovery strategies is a novel delineation of current research that has implications for enhancing agricultural sustainability. By identifying the traits related to P recovery that have been lost to domestication, we can strengthen the P recovery responses by modern crops and reduce P fertilization.

## 1. Introduction

Crop domestication has independently occurred as far back as 12,000 years ago (YA) to as recent as 3000 YA [1,2]. Some of the first crops to be domesticated by early humans were cereals, with wheat (*Triticum aestivum*) being domesticated in Turkey and Syria approximately 10,000 YA [3], barley (*Hordeum vulgare*) in the areas surrounding Iran, India, and Tibet 8000 YA [4], and lentils (*Lens culinaris*) in Southwest Asia between 8000 and 10,000 YA [5]. The outcomes of these domestication events have allowed for increased production and for the incidence of hunger to be reduced [6].

However, crop domestication has also reduced the effective population size of a given crop, as exemplified through domestication bottlenecks [7,8]. For example, only approximately 2% of the ancestral wild population of soybean (*Glycine max*) is estimated to have been used to domesticate early landraces [9]. Similarly, when early tomato (*Solanum lycopersicum*) was transported from Mesoamerica to Europe, there was concomitant sharp decrease in diversity [10]. Continued selection and migration events further diminished tomato genetic diversity [10]. This reduction poses a challenge for production and breeding because of changing environmental conditions and increasing consumer demand paired with a declining pool of genetic information for developing new cultivars [11]. Thus, crop breeders often rely on introgressions from wild relatives to introduce beneficial genes into modern germplasm [12].

Wild crops provide an opportunity to increase abiotic stress resistance in modern crops [12]. In ancient terrestrial ecosystems, phosphorus (P) is limited because of factors such as erosion, weathering, leaching, and low parent material [13,14]. Erosion and weathering cause a gradual disappearance of phosphates from terrestrial soil systems that do not receive regular nutrient additions. In addition to P deficiency, nitrogen (N) is also often deficient and limits production. For plants growing in these older terrestrial ecosystems, low concentrations of N and P are the primary limitations for growth and development [14]. Wild crop relatives are adapted to grow in these nonagricultural soils that have low fertility and thus grow slowly [15]. Conversely, faster growing, cultivated crops are adapted to agricultural soils with high fertility and thus necessitate human intervention for survival [16]. Exogenous fertilizers, for example, are required for modern crops to grow and develop without showcasing symptoms of nutrient deficiency. Approximately 90% of current P fertilization is derived from phosphate rock, a finite mineral [17]. Because P is often the limiting factor for sustaining healthy plant growth, P fertilizers have historically been applied to agroecosystems. With changing agricultural practices from events like the Green Revolution, mining of phosphate rock and global P fertilizer applications have starkly increased [18,19]. In 2022, 42 million Tg P_2_O_5_ were applied globally for agricultural use [20]. This reliance on phosphate rock is notable because the average quality of phosphate rock is decreasing as higher quality reserves are preferentially mined, and current models predict that demand for phosphate rock will continue to increase [21].

Further, although current estimates of total soil P concentrations are above crop requirements (100–3000 mg·kg^−1^), most soil P is not readily available for plant uptake [22,23]. Total estimates can vary with environmental conditions, but on average, only 9% of total soil P is in an available form [24]. When the pool of available P is less than the level required to sustain healthy plant growth, plants must employ strategies to acquire P from sparingly available pools. It has been highly documented that the types of strategies employed by plants varies with domestication [25,26,27,28]. However, a synthesis of this domestication effect for P recovery has not been performed. Here, were explore current literature to elucidate the divergent strategies employed by cultivated crops and their wild relatives to acquire P, including physical and ecological mechanisms. Physical mechanisms are defined those involving root system architecture while ecological mechanisms involve root-microbe associations.

## 2. Phosphorus in Soil and Uptake by Plant Roots

Highly efficient P uptake and transport is important because while P is the 11th most abundant element in the Earth’s crust, the pool of bioavailable P tends to be poor [21]. Depending on the pH of the soil system, bioavailable inorganic orthophosphate exists in different forms: PO_4_^3−^ between pH 10.0 and 14, HPO_4_^2−^ between pH 7.2 and 12.1, and H_2_PO_4_^−^ below pH 7.2 [29,30]. Dihydrogen phosphate (H_2_PO_4_^−^) is the primary form of orthophosphate favored by plants because the maximum uptake of P tends to occurs in the pH range for which this form is dominant (below 7.2) [31] (Figure 1). Inorganic orthophosphate is labile and therefore readily undergoes transformations in the soil. Orthophosphate can be immobilized into organic forms, adsorbed to mineral surfaces, or precipitated into secondary minerals (Figure 1). Secondary P minerals, such as calcium (Ca)-, magnesium (Mg)-, aluminum (Al)- and iron (Fe)-phosphates, are sparingly soluble, and their dissociation depends on soil pH. Iron- and Al-phosphate solubility tends to increase with increasing pH, while Ca- and Mg-phosphate solubility tends to increase with decreasing pH [32] (Figure 1). Primary P minerals (e.g., apatite and strengite) are stable and release P slowly through weathering activity [32]. Through the activity of root exudation, microbial activity, or shifts in root architecture, the solubility of bound-P may increase and become plant available.

Phosphorus (P) constitutes approximately 0.1–0.5% plant dry matter and is an essential plant macronutrient [30,31]. It is a component of nucleic acids and the phospholipid membrane of cells [33]. The primary function of P is as a constituent of adenosine diphosphate (ADP) and adenosine triphosphate (ATP), making it necessary for energy storage and transfer [30,31]. When orthophosphate is cleaved from ADP or ATP, chemical energy is released. ATP is necessary for plant development, as it is a regulator of membrane transport, photosynthesis, and protein biosynthesis. Therefore, when in P deficient conditions, plants show stunted growth [30]. The co-limitation of P and N commonly occurs, and there are strong interactions between the P and N in their signaling pathways [34]. When plants are deficient in P, there is a concomitant decrease in plant N concentration [35]. Further, a component of the Phosphate Starvation Response pathway works in conjunction with the Nitrogen Limitation Adaptation genes to induce responses to P deficiency [34]. Thus, N and P uptake are tightly interconnected and contribute to healthy plant growth.

Unlike the higher concentrations in plant shoot tissue, the concentration of P in soil solution is low, often ranging from 0.001 to 1 mM [30,31]. Further, soil P is immobile, only moving an average of 0.13 mm per day [30]. However, it is important to note that this diffusion coefficient is affected by soil type and water availability; water increases P diffusion, and this effect is increased for lighter-textured compared to clayey soils [36]. Because of this immobility in the soil, plants take up phosphate from the soil surrounding root at a faster rate than phosphate in bulk can translocate to the root zone [37]. Thus, a zone of depletion of available P commonly forms in the rhizosphere [37]. Because of the low concentration and poor mobility of phosphate in soil solution, plants rely on high-affinity root phosphate transporters to uptake phosphate across a steep chemical potential gradient [31]. Phosphorus uptake is an energy-mediated process and requires active symport with H^+^ [32]. There are five phosphate transporter families that moderate P uptake and translocation across different locations within the plant: PHT1 in the plasma membrane drives soil P uptake, PHT2 in the chloroplast inner plastid membrane moderates P translocation, PHT3 in the mitochondrial membrane regulates P distribution, PHT4 in the Golgi apparatus regulates cytosol P transport, and PHT5 in the vacuole regulate vacuolar P transport [38].

## 3. Physical Mechanisms of Soil P Recovery

Here, we define physical mechanisms as those associated with root architecture. Phosphorus is soil-immobile, so there is a heterogenous distribution of P in soil systems [39]. As plants take up P, they create a concentration gradient around the root [40]. As P diffuses through the soil, plant roots intercept this P and take it up through their root system. Because a major method of soil P acquisition is root interception, proper root development to exploit a given volume of soil is essential [41]. Thus, the production of adventitious and ageotropic roots is an advantageous method to scavenge potential pools of soil P [39]. As described in their work with bean (*Phaseolus vulgaris*), Lynch and Brown [42] determined that overarching P efficiency was largely determined by the plasticity in root gravitropism and adventitious root growth. The authors contended that plasticity in root architectural traits was quantitively inherited and can be putatively used in marker aided breeding strategies [42]. Thus, these physical mechanisms of soil P recovery can be used in breeding programs to further enhance the P acquisition efficiency of modern crops. Although both wild and domesticated crops engage in these physical shifts in response to low P, there is a marked difference in the degree to which wild and domesticated crops change their root architecture in response to deplete P conditions.

### 3.1. Topsoil Foraging

Root plasticity expressed through topsoil foraging is a valuable trait which can allow for better exploration of shallow soil horizons [42,43]. Compared to deep soil horizons, surface horizons have greater P concentrations across a range of cropping systems and soil types [24]. This stratification of P is often present in agroecosystems because of manure and fertilizer application, both of which increase soil P concentrations in the top 5 cm of soil over time [44]. Therefore, when P is deficient, there is often a reduction in primary root growth in plants [43]. It is important to consider, however, that this reduction in primary root growth because of P deficiency occurs when P is the most limiting nutrient. Root architectural responses have also been noted for deficiencies of other nutrients such as N and K [45]. The resultant shallower root system can then better exploit a given volume of soil for soil nutrients. This strategy of topsoil foraging has been noted in several horticultural and agronomic crops. Genotypes of maize (*Zea mays*) [46] and bean [42] with shallow root systems have been shown to accumulate more P from soils and express better growth than those genotypes with deeper root systems.

Because of agricultural innovations (e.g., Green Revolution, fertigation), modern agricultural systems have become more exposed to increased total amounts and incidents of fertilization [18,47,48]. Over time, this increased fertilization causes an accumulation of nutrients in the soil surface and therefore a stratification of soil P [44]. Applications of organic material such as compost and manure can also increase the P stratification [49,50]. Current crop cultivars are adapted to grow in these highly fertile and P-stratified environments. Long-term fertilization over the course of centuries may therefore have altered the growth habit of modern crops to promote root growth along the topsoil horizon. For example, Zhao et al. [51] screened over 300 accessions of soybean across a domestication gradient (wild, semi-wild, cultivated) and examined their spatial root configuration under P deficiency. The authors found that under low P, there was a clear evolutionary pathway in root architecture: from deep to shallow root systems [51]. Cultivated soybean had shallow roots, the wild had deep roots, and the semi-wild showed an intermediate root system [51]. The shallow roots system of cultivated soybean allowed it to better acquire nutrients in the topsoil, and modern soybean therefore had greater P efficiency than its wild ancestor.

This trend of root topsoil foraging promotion along a domestication gradient continues for other horticultural and agronomic crops. Akman [52] conducted a study exploring the P efficiency-related root traits of domesticated and wild wheat. In this study, great genotypic diversity was identified, and ultimately, modern wheat varieties were shown to distribute their root biomass along the topsoil region of the soil profile more than wild wheat varieties [52]. Moreover, in their recent comprehensive assessment of P-stressed modern and wild tomato, Demirer et al. [53] illustrated the divergence in primary root elongation. The authors determined that modern tomato showed marked reductions in primary root elongation when P levels were reduced. However, in wild tomato, there was no reduction in primary root growth [53]. Thus, topsoil foraging was more apparent in the modern tomato accession compared to the wild tomato accession. Furthermore, as demonstrated by Rahman et al. [54], domesticated and wild lettuce (*Lactuca sativa*) showed different root architectural traits when exposed to P deficiency. The domesticated variety expressed shallow root growth and enhanced topsoil foraging, whereas wild lettuce showed greater taproot elongation and greater total taproot depth [54]. Therefore, when examining crops across a domestication gradient, there is evidence that domesticated accessions tend to showcase a strategy of reduced primary root growth and enhanced topsoil foraging (Figure 2A).

However, this pattern of decreased primary root length is not as apparent when examining cultivated varieties across time. Ning et al. [80] determined that for maize, newly developed cultivars were shown to possess deeper roots and more root length than previous varieties when they reached the post-silking stage of development. However, prior to maturity and at silking, the newer maize varieties had the same root length as the older varieties. Ultimately, across horticultural and agronomic vegetable crops, there appears to be a trend toward exploiting the topsoil horizon to acquire nutrients as crops progress with domestication, even if select newer cultivars do not follow this trend.

Nutrient-stratified soil environments that are common in agroecosystems provide the ideal environment for the strategy of topsoil foraging to be effective. Conversely, wild crop relatives—which grow in less stratified and more homogeneous natural soils—are not adapted to nutrient stratification and are not equipped to use topsoil foraging as means to recover soil P (Figure 2A). Rubio et al. [55] demonstrated that plants grown in soils with a uniform distribution of P, regardless of concentration, did not benefit from having shallow root systems. Thus, modern crops that have been bred and developed over many generations may be adapted to these nutrient-stratified soils and are therefore better equipped to use topsoil foraging.

### 3.2. Shifts in the Production of Root Types and Density

Plants are sessile organisms, so a necessary strategy to adapt to abiotic stressors (e.g., P deficiency) is to adjust their root spatial configuration [81]. When exposed to mechanical impedance and other stressors, plants may promote lateral rooting by initiating a primordium in the pericycle cells below the root apical meristem [81]. Like topsoil foraging, increasing lateral root density allows for improved P uptake from soil systems as a result of more intensive exploration of soil [82]. Lateral roots are regulated by P availability and play an important role in P solubilization and acquisition because the lateral roots requires less P and biomass investment in development [83]. As shown in greenhouse and field trials with maize, the accessions with greater lateral root density (IBM79, IBM295, IBM301, IBM321) were more efficient in accumulating P [82]. Because of the nutrient stratification that occurs in agroecological soils, lateral rooting may have been positively affected by domestication.

Modern maize and wild teosinte have been shown to display markedly different root structures when grown in low P, likely as a result of different P conditions experienced by wild teosinte and modern maize. For example, domesticated maize readily produced lateral seminal roots whereas wild teosinte did not [26]. This disparity is significant because approximately one third of the P acquired by maize during its life history is acquired through its seminal roots, and thus, domesticated maize is more adapted to acquire P than wild [26]. It has been proposed that the genes for seminal root and lateral root development in modern maize were indirectly selected, but the benefit from these genes is apparent in P acquisition [56]. However, it is important to note that while possessing fewer seminal roots, wild teosinte has an abundance of very fine roots (i.e., <0.03 mm diameter), thus increasing its total surface area to volume ratio [84]. This finding is valuable because root hairs are responsible for up to 50% of total root P uptake [85]. Wild teosinte is adapted to environments with poor P bioavailability, and it may therefore be possible that teosinte utilizes root hairs to acquire soil P.

Although domesticated maize showed greater lateral rooting but less root hair growth compared to wild [26,84], domesticated tomato showed both increased lateral rooting and root hair growth relative to wild [53]. In low P, modern tomato increased lateral rooting and root hair elongation [53] (Figure 2B). Wild tomato was insensitive and maintained consistent lateral rooting and root hair length regardless of P level [53]. Further, greater lateral rooting has been observed in domesticated lettuce rather than wild lettuce [86]. Thus, across different crop species, there may be improvements in rooting strategies to quickly and efficaciously acquire P from fertile soil systems common in agricultural systems, such as through investment in lateral rooting structures.

### 3.3. Selective Partitioning of Biomass and Phosphorus

One method to maximize the potential for root interception of soil P is to preferentially promote root biomass and thereby increase the root-to-shoot ratio (R:S) [39]. At the expense of shoot growth, plants tend to allocate energy and resources into root growth [39,87]. This pattern of selectively partitioning biomass to root tissue rather than shoot tissue in response to P deficiency has been observed in many species such as lantana [87], maize [88], rice [89], and clover [90]. The promotion of R:S is moderated by the P status of the soil. Under conditions of deficient P, sucrolytic activity is largely inhibited in roots, resulting in a greater hexose-to-sucrose ratio, root biomass accumulation, and stable sucrose utilization in roots [88]. Shoot growth is therefore inhibited and carbon is allocated to root tissue [91].

Although there tends to be a relationship between R:S and external P levels, there is variability and inconsistency in reported R:S responses for wild crops and their domesticated counterparts. Araújo et al. [92], across two experiments and two P levels (20 and 80 mg P·kg^−1^ soil), assessed the P efficiency traits of appx. 30 genotypes of common bean. The authors found that in both experiments, regardless of P level, cultivated common bean showed greater R:S than wild varieties [92]. However, certain domesticated agronomic crops show diminished R:S compared to their wild ancestors. Cultivated wheat varieties tend to partition less biomass to their root tissue compared to wild varieties [57]. Similarly, wild teosinte has been shown to have a narrower R:S compared to cultivated maize [84]. These differing responses to P deficiency among species of wild and domesticated crops indicate that the variability in R:S may be genotypic rather than a domestication-driven change.

## 4. Ecological Mechanisms of Soil P Recovery

Ecological mechanisms are defined here as those that are involved in the interactions between plants and their surrounding microbes in the rhizosphere. The rhizosphere is the compartment of soil that is influenced by the root [93]. It is a highly active region comprising diverse microbiota that cohabitate and interact with the host plant [94]. Co-evolved root-associated microbial communities influence the ability of the plant host to tolerate environmental stressors, such as nutrient deprivation [95]. For example, P-solubilizing bacteria transform recalcitrant P to plant-available forms [58] and arbuscular mycorrhizal fungi that help to better exploit a given volume of soil [59] (Figure 2D). These root-microbial associations are essential for plant survival, and it is recognized that plants themselves are not standalone entities, but rather a holobiont of the plant host and its associated microbial communities [96].

Rhizosphere microbial composition can change with environmental conditions and host plant genotype, and the rhizosphere microbiomes of wild crop relatives are often shown to be more sensitive to environmental changes than domesticated varieties [60,61]. These highly responsive microbiomes found in wild crop relatives often vary in composition from the microbiomes of modern crops [28,62,63,64,65]. These shifting microbial communities provide insights into P acquisition strategies and capabilities of the host plant as a result of crop domestication.

### 4.1. Alterations in the Rhizosphere Bacteriome

There are compositional differences between the rhizosphere bacteriome of domesticated and wild crops: wild being enriched in *Bacteroidetes* and domesticated being enriched in *Actinobacteria* and *Proteobacteria* [66,97]. Some of these of the species in these phyla are known to solubilize and mineralize soil P. Phosphorus solubilizing and mineralizing bacteria are able to convert sparingly soluble P into bioavailable forms, principally through the production of extracellular hormones [98]. Fertilizers can affect soil bacterial populations [99], and while the population of some P-solubilizing microbes attenuate in high P soils that are common in agricultural conditions, others maintain their populations in high and low P [100]. These P-solubilizing bacteria can be used as biofertilizers because of their potential to markedly improve P uptake by roots [98,101].

Though both P-solubilizing and P-mineralizing bacteria transform P from sparingly soluble to soluble forms, these two bacterial groups target separate pools of soil P. Phosphorus-mineralizing bacteria target organic P [101]. To mineralize organic sources, P-mineralizing bacteria harbor a member of the pho regulon, a phosphate regulation mechanism involving enzyme and phosphate transporter activity [102]. The most common enzymes capable of mineralizing organic P through the activity of hydrolyzing phosphomonoesters phosphodiesters include alkaline phosphatase (*phoA*), phospholipases (*phoD*), phosphodiesterase (*GlPQ*), and phytase (*PhyC*) [102,103]. The *phoD* gene is highly abundant in soil bacterial populations relative to the abundance of other phosphate-mineralizing enzyme families [104]. This gene is activated by calcium and can effectively hydrolyze both phosphomonoesters and phosphodiesters [104].

Conversely, P solubilizing bacteria do not target phosphoesters, but rather obtain P from poorly soluble precipitated secondary minerals, typically through gluconic acid exudation [105] (Figure 3). For gluconic acid to be produced, the glucose dehydrogenase enzyme and corresponding cofactor, pyrroloquinoline quinone (PQQ) are needed [105] (Figure 3). For PQQ biosynthesis, the pyrroloquinoline quinone synthase C is required, which is encoded by the *pqqC* gene [105], and thus, P-solubilizing bacteria contain *pqqC* (Figure 3).

It has been proposed that wild crop ancestors form stronger associations with the members of their rhizosphere microbiome than modern crops [66,67,68]. In wheat, for example, the functional diversity of the wild ancestor is greater than domesticated relatives [106]. This type of heightened diversity is beneficial because functional traits (e.g., hormone balance, nutrient cycling) promote plant fitness [69]. These microbial differences are further illustrated in the diverging populations of wheat-associated P-cycling bacteria. Wild wheat is more strongly associated with P-decomposing bacteria (targets organic P), while domesticated wheat promotes P-solubilizing bacteria (targets inorganic P) [62]. Agricultural soils tend to have low organic matter compared to native ecosystems, such as grasslands or forests [107]. Thus, the prevalence of these two separate P-cycling bacterial groups in wheat may be a result of the soil environmental factors by which wild and domesticated wheat have evolved.

Further, in tomato, wild relatives were shown to accumulate an abundance of P-solubilizing and P-decomposing bacteria in low-P soil when compared to cultivated tomato [67,68]. These bacteria were also responsive; when P fertilizer was applied, the co-occurrence network complexity increased only in wild tomato, not domesticated [61]. Moreover, Cyanobacteria members have been shown to be enriched in the wild tomato rhizosphere [67], and bacteria in this phylum are important for promoting P cycling [70]. When Cyanobacteria was applied to soils, water-soluble P and Olsen-extractable P increased over a period of seven weeks [71]. Another notable phylum is Bacteroidetes, which tends to be enriched in the rhizospheres of wild crop relatives [57,66]. Bacteroidetes members have marked capacity to break down complex organic compounds, of which many root exudates are composed [57]. Thus, it is possible that this phylum is reliant on root exudates, and therefore, a more plant-responsive microbiome may be present in wild crops.

Although root–bacterial associations may have attenuated post-domestication (Figure 2C), the beneficial bacteria that colonize wild crop relatives are still highly useful for modern cultivars. In potato, it has been shown that inoculation of PSB isolated from its wild progenitor resulted in a significant improvement in biomass and shoot P concentration [108]. Similarly, in rice, P solubilizing bacteria isolated from a wild ancestor was used as an inoculant for modern rice, and researchers found that inoculations improved growth [109].

### 4.2. Associations with Arbuscular Mycorrhizal Fungi

Arbuscular mycorrhizal (AM) fungal symbiosis is estimated to have originated 450 million years ago, and approximately 80% of terrestrial plants form symbiosis with AM fungi [63,110,111], but it has been only in the past 30 years that we have begun to understand the underlying mechanisms of AM fungal symbiosis [63]. There are two main groups of mycorrhizal fungi: aseptate (meaning hyphae are present without separating cell walls, e.g., Glomeromycota) and septate (meaning hyphae are separated by cell walls, e.g., Basidiomycota) [112]. AM fungi are endomycorrhizal fungi because they colonize the intracellular space, rather than the intercellular space and root tip (i.e., ectomycorrhizal fungi (EM fungi)) [112]. While EM fungi are prominent in forest systems, AM fungi are the dominant form of symbiosis on a broader scale [112].

Like high-affinity root phosphate transporters, there are also AM fungi phosphate transporters. The high-affinity (18 μM Km) transporter, GvPT (identified from *Glomus versiforme*) is expressed in the extraradical mycelia, thus indicating its active selection of phosphate from the soil solution. AM fungal transporter utilize either H^+^ (PHO84) or Na^+^ (PHO89) symport, at which point the phosphate enters the arbuscule branch [113]. This process is important because, even though it can be energetically expensive for plants, the colonization of roots by AM fungi can markedly increase P uptake, especially in low-P soils [114].

However, this energetic-nutrient tradeoff does not always net benefit the plant. The carbon cost of AM fungal symbiosis, concomitant with the typical high fertility levels of agricultural soils, indicates that symbiosis with AM fungi in modern plant hosts may be less beneficial for the host plant [63,72,73] (Figure 2D). Current cultivation methods involving high pesticide use, nutrient input, and fallowing further exacerbate AM symbiosis imbalance [63]. While domesticated crops have been shown to benefit from AM fungal colonization, this benefit tends to occur only under P deficiency. For example, Martín-Robles et al. [73] grew 27 varieties of domesticated and wild herbaceous crops representing major families (e.g., Fabaceae, Cucurbitaceae, Poaceae). Upon inoculation of AM fungi in both high P and low P conditions, the representative wild progenitors showed clear benefit, with varieties either increasing biomass or P uptake. Conversely, the domesticated crops only showed increased production when AM fungi were inoculated at an initially deficient concentration of P [73]. Similarly, wild rice is more readily colonized by AM fungi [63]. Even when associating with AM fungi, domesticated rice shows reduced activity of the pathway for mycorrhizal P acquisition [27]. Further, in a field study with wild and cultivated soybean, there was AM fungal symbiosis present in wild soybean, particularly with an enrichment in *Paraglomus*, but no identifiable symbiosis in cultivated varieties [74]. Wild ancestors of sunflower (*Helianthus annuus*), barley, and wheat also form symbiosis with AM fungi more readily than domesticated counterparts [75,76,77].

This pattern of attenuating reliance on AM fungal associations with time continues even after domestication, as shown by wild compared to modern wheat. Varieties of wheat released prior to 1950 show greater dependence on AM fungal symbiosis compared to modern varieties [57,115]. This result indicates that there was apparent AM fungal colonization on wheat varieties released both prior and subsequent to 1950. However, the benefit (through increased P uptake and biomass) was more present in the varieties predating 1950. However, a wheat variety released in the 2000s shows enrichment of Glomeromycota in the rhizosphere, but a weakened association of fungi in general [116]. Therefore, although modern accessions are able to respond to AM fungal associations, especially in low P, their capability to do so at high P levels is diminished compared to wild relatives.

### 4.3. Associations with Non-AM Fungi

Fungal populations are essential to soi health, and fungal richness has been positively related to ecosystem multifunctionality [99,117]. Thus, plants may form associations with both AM- and non-AM fungi as means to acquire soil P [118,119]. For instance, applications of varied fungal isolates to wheat promoted not just bioavailable P concentrations in soils, but also P uptake and plant growth by wheat [120]. In rice, fungal associations have been shown to help promote soil P mobilization in conditions of low P [119]. Further, *Rhizopogon luteolus*, an EM fungus that predominantly colonizes pine tree roots, produces an abundance of extracellular acid phosphatase which targets organic forms of recalcitrant P [118].

Like the diminishing associations with AM fungi, there may also be decreased associations with other fungal populations post-domestication. In wheat, for example, the relative abundance of bacterial and fungal populations changed with domestication [62]. There was a smaller proportionate composition of fungi compared to bacteria in domesticated wheat compared to wild [62]. Wild wheat has also been shown to accumulate more *Ascomycota* [78], a phylum that is known to harbor many different P solubilizing fungal species [79]. Similarly, wild legumes harbor greater diversity of fungal species in the rhizosphere compared to their cultivated counterparts [28]. The fungi associated with wild soybean also possessed more diverse putative beneficial functions compared to cultivated soybean, which harbored fungi capable of organic matter and cellulose decomposition [64]. Wild and cultivated soybean have been shown to recruit different fungi in their rhizosphere, with wild soybean soils being enriched in genera such as *Ascomycota* and *Basidiomycota* and cultivated being enriched in *Funineliformis* and *Rhizophagus* [64,121]. Ultimately, there has been a shift in the composition of fungal populations in the rhizosphere of wild and modern crops, with an apparent reduction in functionality and diversity in domestication varieties.

## 5. Conclusions

Phosphorus (P) is a plant-essential nutrient that showcases poor soil mobility and bioavailability. Wild crop relatives are adapted to grow in unfertilized terrestrial soils, while domesticated cultivars grow in fertile P-stratified agroecosystems. When the pool of bioavailable orthophosphate is low, plants must acquire P from sparingly available sources to grow and develop. While the degree to which P acquisition strategies are employed varies with soil edaphic factors, it also shifts with the host plant at both a genotypic and domestication level. Here, we synthesized evidence from the current literature and showed that there are consistent differences in how domesticated and wild crops interact with the soil to obtain sparingly available P. Modern cultivars rely on direct physical mechanisms to recover soil P and optimize their root architecture to engage in topsoil foraging. This strategy is beneficial to modern crops as agroecosystems tend to showcase nutrient-stratified environments from long-term fertilization. Although modern crops show advantageous root structure, their dialogue with microbial partners is diminished compared to wild species. Across species, wild crop relatives emphasize an ecological approach to form more intimate associations with bacterial and fungal communities. This novel conclusion that wild relatives consistently promote an ecological response while domesticated crops engage in physical mechanisms to acquire soil P provides a valuable insight into how we can promote soil P cycling in modern cultivars. By understanding what mechanisms were lost to domestication, we can begin to develop methods to reincorporate these strategies and reduce exogenous P inputs in agricultural systems.

## Figures and Tables

**Figure 1 plants-14-02296-f001:**
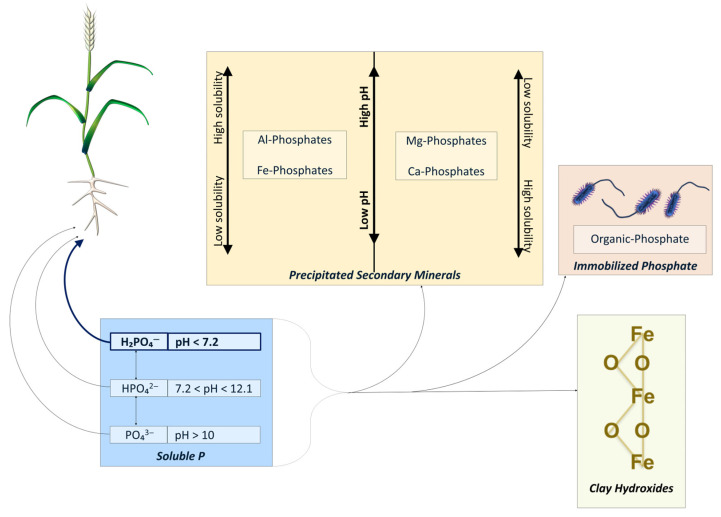
Common forms of loss of soluble phosphorus (P) from soil solution. The form of soluble P varies with pH (blue box). Soluble P is plant available and is taken up by roots. H_2_PO_4_^−^ is the predominant form of phosphate taken up by plant roots (bolded text in blue box). Soluble P can react with cations to form precipitated secondary minerals (dark yellow box). The solubility of the P forms varies with pH (indicated by the arrows in the dark yellow box). Soluble P can also be immobilized to organic forms (red-orange box) or fixed to clay matrices (light yellow box). Figure generated by the authors with information culled from sources within the article [29,30,31,32].

**Figure 2 plants-14-02296-f002:**
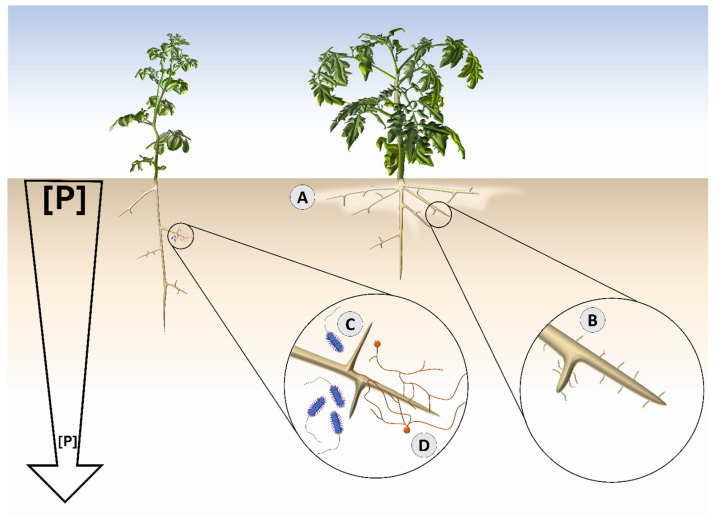
Prototypical methods of soil phosphorus (P) acquisition in P deplete conditions in wild (left) and modern (right) crops. Relative soil P concentration is illustrated with the arrow on the left (topsoil with high concentration and deeper soil with low concentration). To recover P in low-P soils, modern crops tend to exploit topsoil horizons which have a greater abundance of available P compared to lower soil horizons (A) and show responsive root hair growth that increases in length with decreasing concentration of P (B). Wild crop relatives show strong responsive associations with soil P solubilizing bacteria (C) and with arbuscular mycorrhizal fungi (D). Figure generated by the authors using information culled from sources cited within this article [18,26,27,28,44,47,48,49,50,51,52,53,54,55,56,57,58,59,60,61,62,63,64,65,66,67,68,69,70,71,72,73,74,75,76,77,78,79].

**Figure 3 plants-14-02296-f003:**
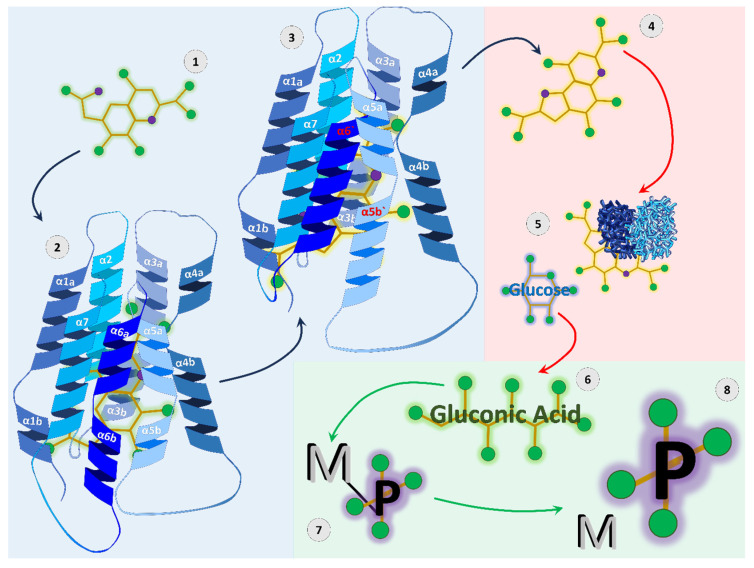
Mechanism of phosphorus (P) solubilization by the *pqqC* gene. The *pqqC* gene regulates soil P solubilization through biosynthesis of pyrroloquinoline quinone (PQQ) (blue background), subsequent catalyzation of gluconic acid (red background), and dissociation of metal-phosphate compounds (green background). An intermediate of PQQ enters the 7-helix structure of *pqqC* (steps 1 and 2). Once in the reaction matrix, *pqqC* undergoes conformational change at the α5b and α6 helices (step 3). PQQ is then released from the reaction matrix (step 4). PQQ functions as a cofactor to glucose dehydrogenase and catalyzes the reaction to form gluconic acid initially from glucose (steps 5 and 6). The acidification and reductant potential of gluconic acid allows it to react with metal-phosphate compounds (step 7). The metal-phosphate compounds dissociate, and bioavailable phosphate is released into the soil solution (step 8). Figure generated by the authors from information cited within this article [98,105].

## Data Availability

No new data were created or analyzed in this study.

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
