# Peer review of "Assessing the Divergent Soil Phosphorus Recovery Strategies in Domesticated and Wild Crops"

_plants, 2025, doi:10.3390/plants14152296_

Round 1

Reviewer 1 Report

Comments and Suggestions for Authors

Very basic information is given at the beginning. Not all the data are needed here. The article is just a literature review. It could have been a chapter in a textbook on crop fundamentals. The conclusions do not contain any new information. The information provided is common knowledge. The paper contains some editorial errors. 7 comments are marked in the text. I believe that this is not a suitable article for inclusion in a reputable journal.

Reviewer 2 Report

Comments and Suggestions for Authors

ID:plants-3735711

Comments:

  1. Lines 8-9: Phosphorus fertilizers are produced by the chemical transformation of insoluble phosphates into soluble phosphates.
  2. Lines 42-43: Why erosion; the reader expects an explanation.
  3. Lines 43-45: What about nitrogen in these habitats?!
  4. Lines 49-51: It started much earlier. Liebig's Law in its sense concerned phosphorus?!
  5. Lines 74-75: Without energy in the process of nutrient uptake, including the available nitrate N, there is no growth and therefore no plant yield. It is this form of N that dynamizes plant growth; therefore, the reader expects a few words about the interaction of both of these nutrients.
  6. Lines 81-84: However, this whole mechanism (plant growth rate and P demand) is driven by nitrate nitrogen availability in the soil and it uptake. Here, the authors have to reach for the nutrient diffusion coefficients in the soil. It should not be forgotten that the dynamics of nutrient uptake is controlled by the water content in the soil.
  7. Lines 91-106: This part should start this section. A diagram for these dependencies should be provided or discussed (Lindsay, 1979).
  8. Lines 111: „root interception”?! Research shows that the dominant mechanism is diffusion!!!! It is advisable for the authors to familiarize themselves with a wider pool of scientific materials in this field.
  9. Lines: 130-131: It's not that simple. It depends on the interaction with nitrogen and potassium, and so on.
  10. Lines 133-135: And which variant had a greater yield?
  11. Lines 139-140: Stratification in the soil is determined by the humus layer.
  12. Line 142, Figure 1A: The discussed examples of responses of different ecotypes of crop plants should be discussed in the text and not only in the figure description (footnote).
  13. Lines 199-202: And which variant had a greater yield?
  14. Lines 203-213: The growth environment of wild teosinte is extremely poor in P, which forces the plant to use a specific strategy to obtain P.
  15. Line: 230: „Undercondition of low”, but not deficient P?!….
  16. Line 242: „a greater R:S”ratio” or perhaps it should be more correctly „narrower R:S ratio?!
  17. Lines 245-260: Without analysis of the interaction of P with N, this whole argument is only seemingly true. The greater the plant biomass, including the greater the P dilution effect, the greater the PPUE. The driver of plant growth is nitrogen. The analysis of this indicator, i.e. PPUE, is typically mechanistic.
  18. Lines 267-269: Figure 1D should be referenced at this point.
  19. Lines 375-376; 386-388: In an article like this, sometimes it is necessary to try to explain the phenomenon being described to the reader.
  20. The conclusions are not really conclusions, but merely a summary.

Final conclusion: The authors should always associate such type of scientific material with management of N by crop plants. Without taking into account the interactions of N and P, this whole analysis is so-called art for art's sake.

Reviewer 3 Report

Comments and Suggestions for Authors

Dear colleagues
I have studied your paper,
in general it is written very readably and even a reader with low knowledge has no problem understanding it - that is completely fine

perhaps it would be possible to indicate the conclusions of the article more in the abstract - one or two sentences are enough.

The paper itself is a compilation of high-quality and up-to-date sources - I have no major comments, of course it is true that this type of paper is harder to oppose...

Round 2

Reviewer 1 Report

Comments and Suggestions for Authors

The article has changed very little. The article is still just a literature review. It could have been a chapter in a textbook on crop basics. The conclusions have been improved. The paper still contains editorial errors. 4 comments have been marked in the text.
